# Butyric Acid Production by Fermentation: Employing Potential of the Novel *Clostridium tyrobutyricum* Strain NRRL 67062

Nasib Qureshi [1],*, Siqing Liu [2] and Badal C. Saha [1]

1   United States Department of Agriculture (USDA), Agricultural Research Service (ARS), National Center for Agricultural Utilization Research (NCAUR), Bioenergy Research Unit, 1815 N University Street, Peoria, IL 61604, USA
2   USDA, ARS, NCAUR, Renewable Products Technology Research Unit, 1815 N University Street, Peoria, IL 61604, USA
*   Correspondence: nasib.qureshi@usda.gov; Tel.: +1-(309)-681-6318; Fax: +1-(309)-681-6427

**Abstract:** In this study, the ability of a novel strain of *Clostridium tyrobutyricum* NRRL 67062 to produce butyric acid during glucose fermentation was evaluated. The strain was evaluated for substrate and product inhibition in batch experiments using anaerobic tubes. To characterize glucose inhibition, initial glucose concentrations ranging from 60 to 250 g L$^{-1}$ were used, and it was demonstrated that a glucose concentration of 250 g L$^{-1}$ exerted strong inhibition on cell growth and fermentation. To evaluate butyric acid inhibition, the culture was challenged with 5–50 g L$^{-1}$ of butyric acid at an initial pH of 6.5. These experiments were performed without pH control. When challenged with a butyric acid concentration of 50 g L$^{-1}$, cell growth was slow; however, it produced 8.25 g L$^{-1}$ of butyric acid. This suggested that the butyric acid tolerance of the culture was 58 g L$^{-1}$. In a scaled-up batch experiment, which was performed in a 2.5 L fermentor with an initial glucose concentration of 100 g L$^{-1}$, the pH was controlled at 6.5. In this experiment, the strain produced 57.86 g L$^{-1}$ of butyric acid and 12.88 g L$^{-1}$ of acetic acid, thus producing 70.74 g L$^{-1}$ of total acids with a productivity of 0.69 g·L$^{-1}$·h$^{-1}$. A concentration of 70.74 g L$^{-1}$ of acids equates to a yield of 0.71 g of acid per g consumed glucose. The maximum cell concentration was 3.80 g L$^{-1}$, which may have been the reason for high productivity in the batch culture. Finally, corn steep liquor (CSL; a commercial nutrient solution) provided greater growth and acid production than the refined medium.

**Keywords:** butyric acid; sugar inhibition; *Clostridium tyrobutyricum*; productivity; yield; corn steep liquor (CSL)

## 1. Introduction

With accelerating climate change, there is increasing interest in developing new sources of sustainable chemicals and fuels. Ethanol is already produced by fermentation, and other candidates for biochemical production include butanol, isobutanol, and acetic, butyric, succinic, and anthranilic acids [1–6]. Butyric acid, in particular, can be directly produced by fermentation and used in the manufacturing of pharmaceuticals, perfumes, plastics, plasticizers, and fuels, as well as in the making of butyrate esters, such as cellulose butyrate [4–7]. It can be converted into important fine chemicals and biofuels such as butanol and butyl–butyrate [8,9]. Butyric acid can be produced from numerous sources of sugars including glucose derived from corn, sugarcane molasses, beet molasses, paper mill sludges, sweet sorghum stalks and juice, and hydrolysates of cellulosic residues including straws (wheat, barley, and rice), reed canary grass, and switchgrass [7,10–13]. Cellulosic feedstocks require converting plant cell wall glucans to glucose prior to fermentation.

Butyric acid can be produced by fermentation by using *Clostridium tyrobutyricum*. Recently, we reported on the strain *Clostridium tyrobutyricum* NRRL 67062, which was isolated by our laboratory [7]. The strain looked like a promising candidate for butyric acid production based on preliminary results. It was capable of producing 9.87 g L$^{-1}$ of

butyric acid from corn stover hydrolysate. However, this process used a refined laboratory medium at a high concentration, which is undesirable for commercial production because of the cost. The end titer is also too low for economic recovery. The traditional recovery method of butyric acid from fermentation broth is distillation. However, the boiling point of butyric acid is much higher (163.5 °C) than water (100 °C), so it requires more energy to separate from fermentation broth. To combat energy inefficient recovery, novel efforts are being made to recover butyric acid from fermentation broth [14–19].

In addition to the application of energy efficient product recovery techniques, the economic potential of this process can only be realized if followings are considered: (i) the use of cost-effective nutrient media and (ii) high-productivity reactors. There are several kinds of nutrient media that can be applied to industrial fermentations in general, including reinforced clostridial medium (RCM), Luria–Bertani medium (LBM) [20], P2 medium [21,22], trypticase soy broth (TSB), and corn steep liquor (CSL) [23–25]. RCM, LBM and P2 media are refined and costly, so their application may not be cost-effective for this large-volume/low-value product, thus leaving TSB and CSL as the possible viable options.

In this study, *Clostridium tyrobutyricum* NRRL 67062 was evaluated for further properties that are essential for it to be considered for industrial use. The studied properties included its robustness to high sugar and butyric acid concentrations, both of which are needed to achieve high butyric acid titers. The strain was further evaluated for fermentations with lower nutrient additions, and the refined nutrient as replaced with an inexpensive unrefined nutrient source commonly used for industrial fermentations (e.g., CSL). Finally, the bacterium was evaluated in a pH-controlled bioreactor to determine the maximum obtainable butyric acid production from the fermentation of glucose. Lastly, this study allows for the performance of this strain to be more precisely compared with results reported for other strains considered for butyric acid production.

## 2. Materials and Methods

### 2.1. Micro-Organism and Culture Propagation

*C. tyrobutyricum* NRRL 67062 was previously isolated and deposited in ARS Culture Collection (Peoria, IL, USA; https://nrrl.ncaur.usda.gov/, 22 July 2022). Spores of the culture were developed in a reinforced clostridial medium (RCM, 38 g·L$^{-1}$; Difco$^{TM}$, Becton, Dickinson and Company, Sparks, MD, USA) supplemented with a filter-sterilized glucose (60 g L$^{-1}$; Fisher Scientific, Fair Lawn, NJ, USA) solution, followed by their storage in the fermentation broth at 4 °C. In order to prepare the liquid RCM, 19 g of solids was dissolved in 500 mL of distilled water, followed by the transfer of the medium to a 1 L Pyrex$^{TM}$ screw cap glass bottle and sterilization at 121 °C for 15 min. The liquid RCM was stored at 4 °C for further use. A glucose solution (Fisher Scientific; 400 g L$^{-1}$) was prepared and filter-sterilized by passing it through a 0.20 µm filter (Whatman, Maidstone, UK). This solution was stored at 4 °C to prepare the fermentation medium for cell growth and butyric acid production.

In order to prepare the medium for culture propagation, 5 mL of the RCM and 250 µL of a glucose solution were transferred to a 15 mL polypropylene culture tube (Fisherbrand, Fisher Scientific). This solution was sparged with oxygen-free nitrogen gas at a flow rate of 5–10 mL·min$^{-1}$ for 7 min. Foaming was often observed during sparging, which was controlled by 100 × diluted sterile antifoam (Antifoam 204, Sigma Chemicals, St. Louis, MO, USA). Then, the tube containing the liquid medium was transferred to an anaerobic jar (BBL GasPak$^{TM}$, Sparks, MD, USA) and incubated for 48 h at ambient temperature. The anaerobic condition was maintained by using BD GasPak$^{TM}$ EZ envelopes (Sigma Chemicals). The spore broth (300 µL) was heat-shocked on a heating block (Cole-Parmer$^{TM}$, Vernon Hills, IL, USA) at 75 °C for 2 min, and then we transferred 100 µL of spores to the propagation tube. The tube was placed in an anaerobic jar (BBL GasPak$^{TM}$) and incubated at 35 °C for approximately 24 h or until cell growth was observed. This was inoculum I and had an optical density (OD) of 0.53 at a wavelength (λ) of 540.

To prepare inoculum II, 26.25 mL RCM solution and 8.75 mL filter-sterilized glucose solution were transferred to a 50 mL sterile polypropylene tube (Fisherbrand, Fisher Scientific). Prior to inoculation, anaerobic environment was created by sparging 50 mL nitrogen gas·min$^{-1}$ for 10 min. After sparging with gas, the tube containing the medium was also placed in an anaerobic jar (BBL GasPak$^{TM}$) for 48 h at ambient temperature. Upon inoculation with 3 mL of the abovementioned inoculum, the tube was placed in an anaerobic jar (BBL GasPak$^{TM}$), which was incubated at 35 °C for 12 h (inoculum II; OD 0.94). This culture was used to inoculate the tube cultures. For the inoculum used in the bioreactor, 3.8 g of the RCM was dissolved in 75 mL of distilled water in a 125 mL screw-capped glass bottle and autoclaved at 121 °C for 15 min. A glucose stock (3 g of glucose in 15 mL of distilled water) was separately autoclaved and added to the RCM contained in the bottle. The bottle culture was made anaerobic as described above. The bottle was inoculated with 10 mL of the inoculum developed above and incubated at 35 °C for 12 h. This was inoculum III with an OD of 0.93.

### 2.2. Glucose and Butyric Acid Inhibition Studies

For sugar inhibition studies, various amounts of presterilized RCM and filter-sterilized 400 g L$^{-1}$ glucose solution were transferred to a 15 mL polypropylene tube (Fisherbrand$^{TM}$, Fisher Scientific). The total volume in each tube was 9.33 mL. Glucose concentrations ranged from 60 to 250 g L$^{-1}$. For the butyric acid inhibition study, 7.00 mL of RCM (Difco$^{TM}$) and 2.33 mL of the glucose solution were transferred. Prior to developing the anaerobic condition, butyric acid was added (from 99% pure solution; Sigma Chemicals) to the tubes followed by adjustment of pH to 6.5. In the tubes, anaerobic condition was developed by sparging 10 mL min$^{-1}$ oxygen-free nitrogen gas for 6 min. Additionally, the tubes with loose screw caps were put in an anaerobic jar (BBL GasPak$^{TM}$) for 48 h at 25 °C. One milliliter samples were taken from the tubes and the reactor, and then we centrifuged them (Eppendorf 5417C, Hamburg, Germany) at 12,000 rpm for 3 min. Prior to centrifuging, OD was measured at 540 nm using a spectrophotometer (DU 800; Beckman Coulter, Inc., Fullerton, CA, USA) as described in the Section 2.5. The clear supernatant was stored at −18 °C until it was analyzed for residual glucose, acetic acid, and butyric acid.

### 2.3. Fermentation with Corn Steep Liquor (CSL)

In order to compare studies with the RCM, 38 g L$^{-1}$ CSL (49% dried solids; Sigma Chemicals) solution was prepared using distilled water and autoclaved at 121 °C for 15 min. The solution was cooled to room temperature and stored at 4 °C for experiments. To prepare the medium, various amounts of CSL (Sigma Chemicals) were added to the tube (1.9, 4.7, 9.3, 14.0, and 18.6 g L$^{-1}$; dried solids). In all tubes, the glucose (Fisher Scientific) concentration was kept constant at 100 g L$^{-1}$. The initial pH was adjusted to 6.5 with 5 M NaOH (EMD Chemicals Inc., Gibbstown, NJ, USA). Anaerobic conditions were developed as described above for the sugar and butyric acid inhibition experiments.

### 2.4. Butyric Acid Fermentation Studies in Scaled-Up Reactor

The butyric acid production was scaled up to a 2.5 L reactor (Bioflo 2000 Fermentor, New Brunswick, NJ, USA). To prepare 1 L of media, 38 g of RCM was dissolved in 750 mL of distilled water and transferred to the bioreactor. Then, 250 mL glucose (Fisher Scientific) solution containing 100 g of glucose was prepared and separately autoclaved. Upon cooling, the glucose solution was added to the reactor. To create anaerobic conditions, the medium was sparged with oxygen-free nitrogen gas at a flow rate of 1 vvm (vol. per vol. per min.) and agitation at 150 rpm (revolution per min) for 5 h, after which nitrogen gas sweeping across the medium surface was started at a flow rate of 50 mL per min. The pH probe (Hamilton EasyFerm Bio HB S8, 225 mm; Reno, NV, USA) was sterilized using 50% (*v/v*) ethanol for 2 h, and then it was washed with sterile distilled water and inserted into the reactor. pH was automatically controlled at 6.5 using 5 M NaOH. The incubation temperature of the reactor was controlled at 35 °C. Both of these conditions were optimized

prior to these experiments. Since optimization procedures of fermentation conditions are basic and simple in nature, it was decided not to include them in this article.

*2.5. Analyses*

Acetic and butyric acids were measured with gas chromatography (GC; 6890N, Agilent Technologies, Wilmington, DE, USA). Analytical-grade standard acetic (Sigma Chemicals) and butyric (Sigma-Aldrich, St. Louis, MO, USA) acids were obtained from these companies. Details of their measurement have been published elsewhere [21,22]. Prior to injecting the samples to the GC, they were diluted by 20 fold using distilled water. Glucose was measured with HPLC (Thermo-Fisher Scientific, Pittsburgh, PA, USA). Experimental samples were diluted by 20 fold with distilled water, centrifuged for 20 min at 12,000 rpm using an Eppendorf 5417C centrifuge, and filtered through 0.20 μm filter (Cole-Parmer, Vernon Hills, IL, USA) before being injected into HPLC column (Bio-Rad 87P, 300 mm; Hercules, CA, USA). A Bio-Rad Micro-Guard De-Ashing cartridge and a Bio-Rad Micro-Guard Carbo-P cartridge were also used. The solvent used was MilliQ water at a flow rate of 0.6 mL·min$^{-1}$. The column temperature was maintained at 75 °C. Cell density was measured with a spectrophotometer (DU 800; Beckman Coulter). Before measuring optical density, the samples were diluted by 10–20 fold using a 9 g L$^{-1}$ saline (NaCl; Sigma Chemicals) solution. Blank sample contained a 9 g L$^{-1}$ saline solution. Cell concentration was determined using a graph plotted between optical density and cell dry weight concentration, and it is reported as dry weight in g L$^{-1}$. The total acid productivity was defined as the total acid produced in g L$^{-1}$ divided by fermentation time and is reported as g L$^{-1}$ h$^{-1}$. The specific productivity is defined as productivity divided by cell concentration. Acid and cell yields are reported as total acid or cell concentration divided by glucose used and are presented as g g$^{-1}$. Although there may have been other chemicals present in the fermentation mixture that were below the detection limit of gas chromatography, glucose utilization material balance was performed using butyric and acetic acids as the final products. The results presented in this article are an average of two replications, with error margins of ±2 to 9%. The authors confirm that under the laboratory conditions mentioned in the Section 2, these results are reliable and the experiments are repeatable.

**3. Results and Discussion**

*3.1. Control Experiment and Substrate Inhibition*

A control fermentation was run using standard laboratory nutrients (RCM) and glucose (60 g L$^{-1}$). The fermentation produced 15.62 g·L$^{-1}$ of butyric acid and 5.29 g L$^{-1}$ of acetic acid, thus totaling 20.90 g L$^{-1}$ of acids (Figure 1A). All the glucose was consumed after 42 h, and the product productivity was 0.50 g L$^{-1}$ h$^{-1}$. In the fermentation broth, a final cell concentration of 2.50 g L$^{-1}$ was measured with a specific productivity of 0.20 h$^{-1}$. Since the 60 g L$^{-1}$ of glucose was completely utilized, an experiment was performed to find out the maximum glucose concentration that this strain could tolerate. Fermentations were run at glucose concentrations ranging from 100 to 250 g Lg L$^{-1}$. A glucose concentration of 100 g L$^{-1}$ produced a butyric acid concentration of 13.32 g L$^{-1}$ and an acetic acid concentration of 4.37 g L$^{-1}$. The culture was actually able to grow with 250 g L$^{-1}$ of glucose (Figure 1B); however, it only produced 5.60 g·L$^{-1}$ of butyric acid and 2.91 g L$^{-1}$ of acetic acid. At initial glucose concentrations of 100, 150, 200, and 250 g L$^{-1}$, cell titers reached 1.62, 1.38, 1.14, and 1.40 g L$^{-1}$, respectively. Total acid productivities, specific productivities, acid yield, and cell yield are presented in Figure 1B. The residual glucose concentrations were 0–200 g L$^{-1}$ (Figure 1C). The end product titer and productivity were highest for the 60 g L$^{-1}$ glucose fermentation. The product yield was highest for the 100 g L$^{-1}$ glucose fermentation when residual glucose was taken into account. Interestingly, the specific productivity was constant up to 150 g L$^{-1}$ of glucose, indicating that decreased product yields reflected poorer cell growth. The lower specific productivities at higher sugar concentrations probably reflected osmotic stress. Medium components such as salts [26] and sugars [20,27] are known to cause osmotic stress in microorganisms. Since our medium (RCM) did not have more than 3.0 g L$^{-1}$ of sodium salt, the effect of osmotic pressure due to salt was disregarded.

However, osmotic pressure due to increased sugar concentrations (100–250 g L$^{-1}$) may have played a major role in reducing the levels of butyric and acetic acid production. It is known that osmotic shock caused by, inoculation of cells into a high-sugar-concentration solution may modify the phospholipid structure of the cell membranes or may even kill cells.

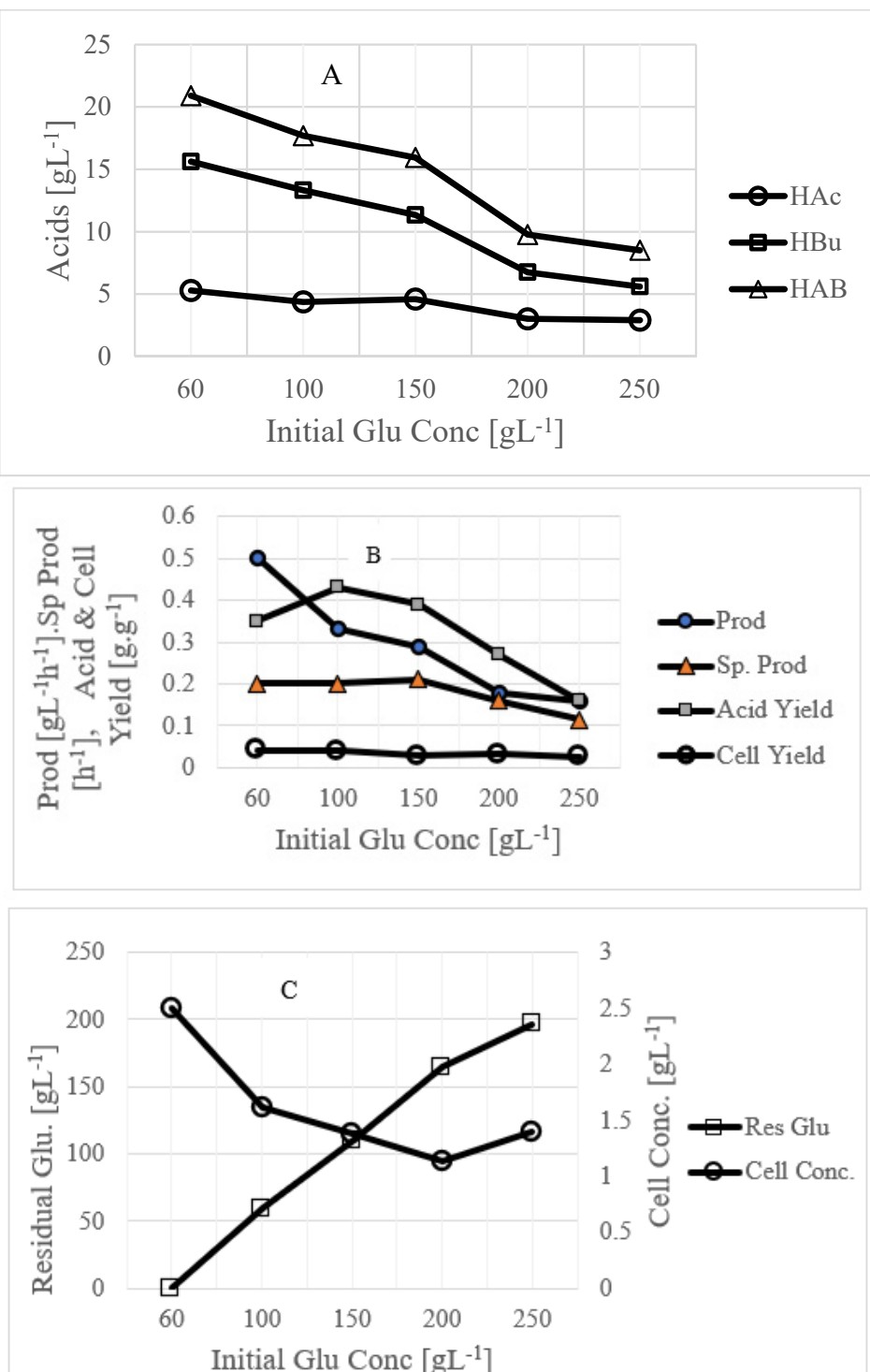

**Figure 1.** Glucose inhibition on *C. tyrobutyricum.* (**A**) Production of acids at various glucose concentrations; (**B**) productivities, specific productivities, acid yield, and cell yield; (**C**) residual glucose and cell concentration. HAc—acetic acid; HBu—butyric acid; HAB—total acetic and butyric acids; Prod—productivity; Sp. Prod—specific productivity; Res Glu—residual glucose; cell conc—cell concentration.

### 3.2. Butyric Acid Inhibition

Next, an experiment was performed to determine butyric acid inhibition. Prior to inoculation, various amounts of butyric acid (from 5 to 50 g L$^{-1}$) were added to the medium (Figure 2A). The cultures produced 9.57 to 15.23 g L$^{-1}$ of total acids on top of the added or challenged butyric acid, thus suggesting that this strain can tolerate and produce more than 50 g L$^{-1}$ of butyric acid. To our knowledge, there have been no butyric acid inhibition studies (similar to ours) in the literature using other strains such as *C. tyrobutyricum* ATCC 25755 [10,17], *C. tyrobutyricum* DSMZ 2637 [16], and *C. thermobutyricum* JW171K [13], which makes it difficult to compare our results. However, the maximum production levels of these strains have been compared and are presented in the following paragraphs. Figure 2B shows cell concentrations and residual glucose levels for our strain. Up to an added butyric acid concentration of 25 g L$^{-1}$, cell growth was not inhibited and cultures reached cell concentrations of 2.0–2.08 g L$^{-1}$. Challenging the culture with more than 25 g L$^{-1}$ of butyric acid sharply reduced cell growth. For example, when 30 g·L$^{-1}$ of butyric acid was added, the final cell concentration was only 1.15 g L$^{-1}$. It was noted that residual glucose levels were also decreased. During fermentation, glucose is used for cell growth, cell maintenance, and product formation. It is likely that more glucose was used by the cell for cell maintenance [28] and was therefore the reason for decreased sugar levels. Figure 2C shows the productivities, specific productivities, total acid yield, and cell yield. In these experiments, the total acid yields were from 0.32 to 0.51 g g$^{-1}$. Microorganisms experience acid stress during the bioproduction of acids [29]. The increased production of acids adversely affects membrane integrity and fluidity, and hence metabolic regulation. As a result, at a threshold concentration of acids, cell growth and fermentation are arrested.

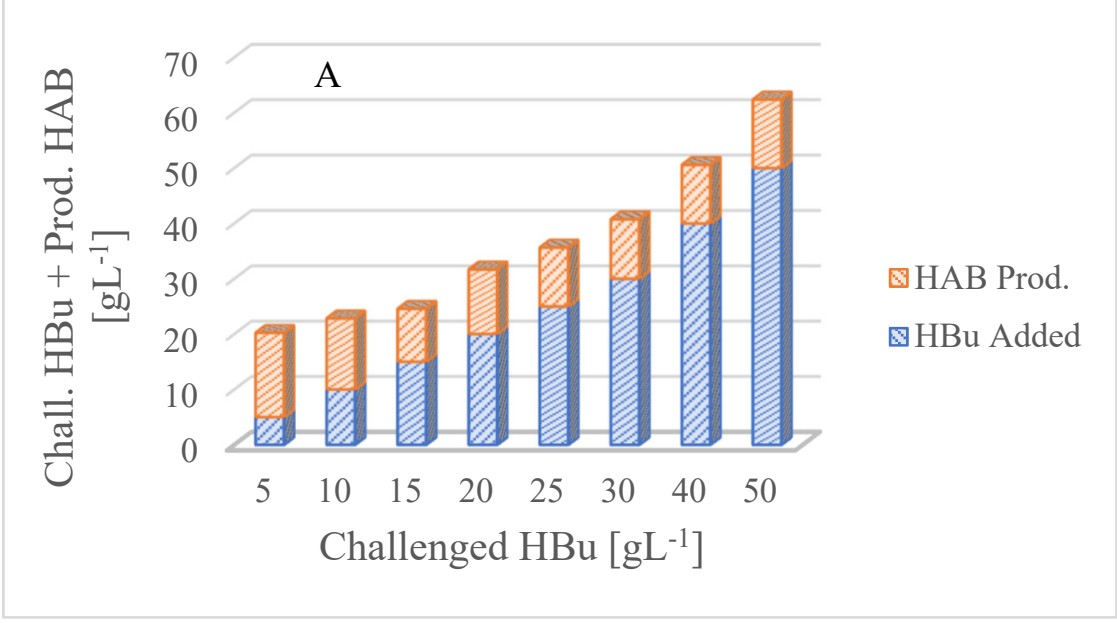

**Figure 2.** *Cont.*

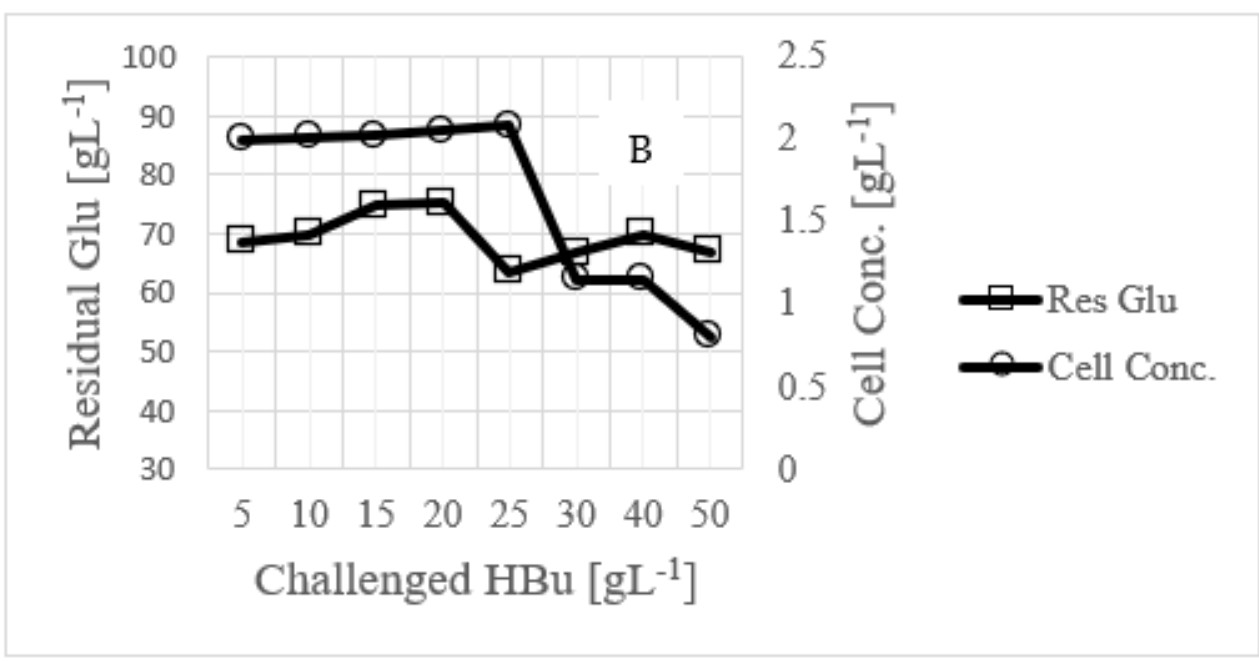

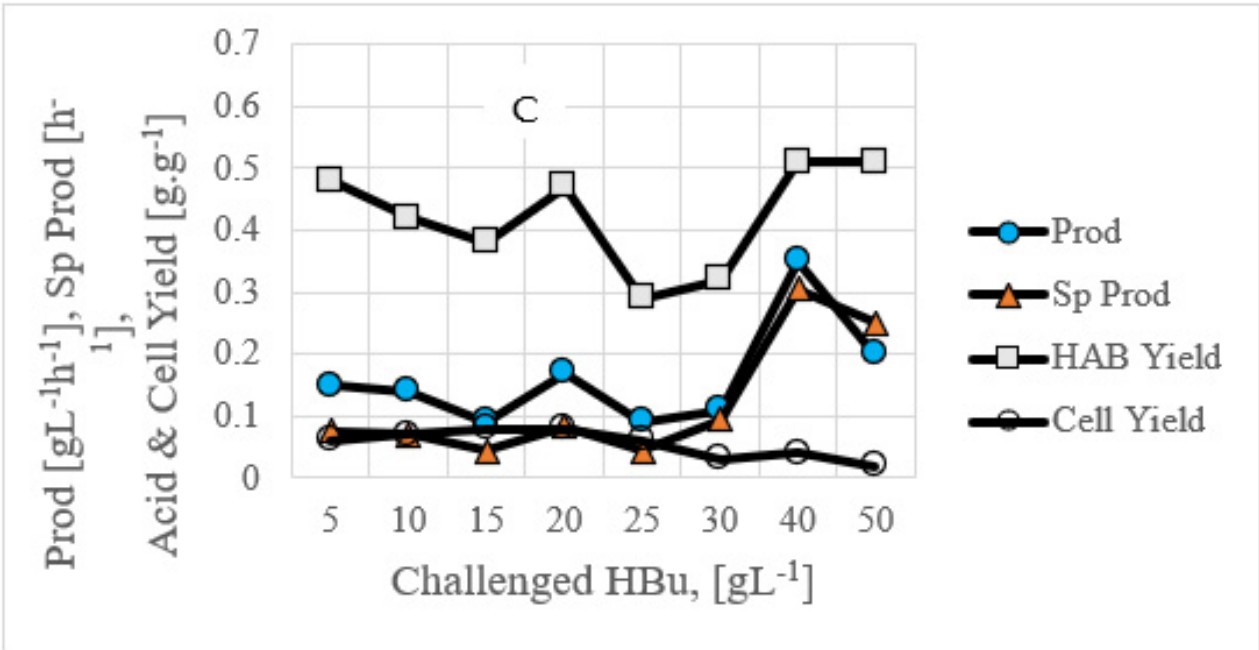

**Figure 2.** Butyric acid inhibition on *C. tyrobutyricum*. (**A**) Production of acids at various added butyric acid concentrations; (**B**) residual glucose and cell concentration; (**C**) productivities, specific productivities, acid yield, and cell yield.

### 3.3. Use of Commercial Nutrient Medium Such as CSL

The abovementioned experiments used a standard RCM concentration of 38 g L$^{-1}$, which is only cost-effective for laboratory studies. Hence, we attempted to lower the amount of the RCM by testing concentrations from 3.8 to 38.0 g L$^{-1}$ (Figure 3A). Even lowering the RCM concentration from 38 to 28.5 g·L$^{-1}$ significantly lowered the fermentation yield. At the lowest tested RCM concentration (3.80 g L$^{-1}$), the maximum achieved concentrations were only 3.41 g L$^{-1}$ of butyric acid and 2.71 g L$^{-1}$ of acetic acid. The residual glucose and cell concentrations are shown in Figure 3B. Residual glucose concentrations were from 70.9 to 88.6 g L$^{-1}$. This experiment demonstrated that a reduction in the RCM concentration was not beneficial for this fermentation. In these experiments, butyric acid productivities were 0.13 to

0.26 g $L^{-1}$ $h^{-1}$ and specific productivities were 0.16–0.59 $h^{-1}$. The total acid yield and cell yield were 0.34–0.53 and 0.02–0.05 g $g^{-1}$, respectively (Figure 3C).

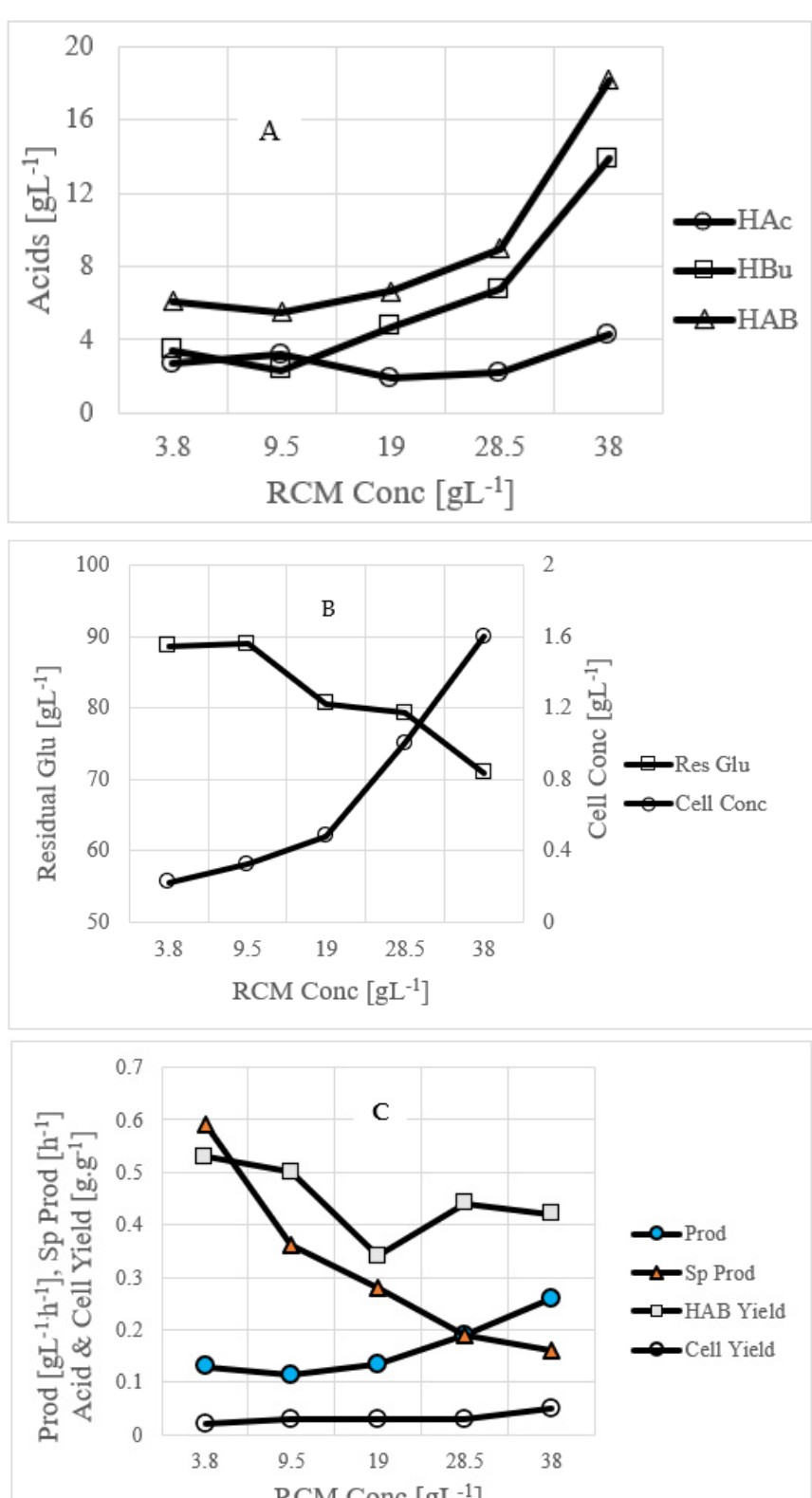

**Figure 3.** Effect of RCM on butyric acid production by *C. tyrobutyricum*. (**A**) Production of acids at various RCM concentrations; (**B**) residual glucose and cell concentration; (**C**) productivities, specific productivities, acid yield and cell yield. RCM—reinforced clostridial medium.

Since low RCM concentrations are not favorable for the production of high concentrations of butyric acid, we explored substituting CSL, which is a commercial source of nutrients widely used in the fermentation industry. CSL is manufactured as a co-product from condensed steep. It is favored by the industry because it is a rich source of nutrients and inexpensive. Two other evaluated commercial nutrient sources, trypticase soy broth (TSP) and P2 (a semi-defined medium commonly used for clostridium fermentations), were found to provide inferior results (data not shown) compared with the use of CSL. Between 1.90 and 18.6 g L$^{-1}$ (dry weight) of CSL was added. At a CSL concentration of 18.6 g L$^{-1}$, 3.86 g L$^{-1}$ of acetic acid and 11.57 g L$^{-1}$ of butyric acid were produced (Figure 4A). The final cell growth was 2.6 g L$^{-1}$, which was higher than that obtained with the RCM. Residual glucose concentrations were from 54.9 to 80.7 g L$^{-1}$ (Figure 4B). In this fermentation, the productivity ranged from 0.19 to 0.42 g L$^{-1}$ h$^{-1}$ and specific productivity ranged from 0.15 to 0.49 h$^{-1}$. The total acid yields were from 0.34 to 0.50 g g$^{-1}$, and cell yields were 0.02–0.08 g g$^{-1}$ (Figure 4C). CSL is a rich source of degradable proteins, amino acids, peptides, trace elements, and other nutrients. It is anticipated that due to the abovementioned nutritive value, CSL resulted in successful butyric acid fermentation. The use of CSL has been reported as a cost-effective nutrient source for ethanol [24,25] and *Bacillus subtilis* [23] fermentations.

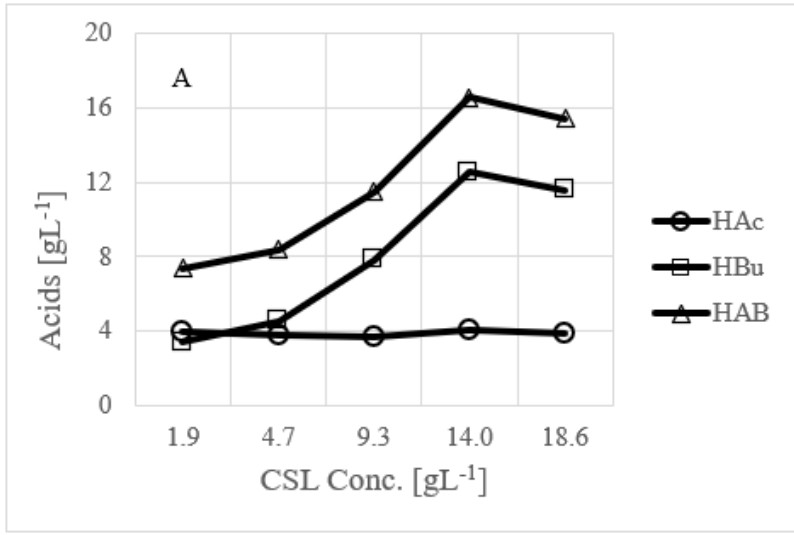

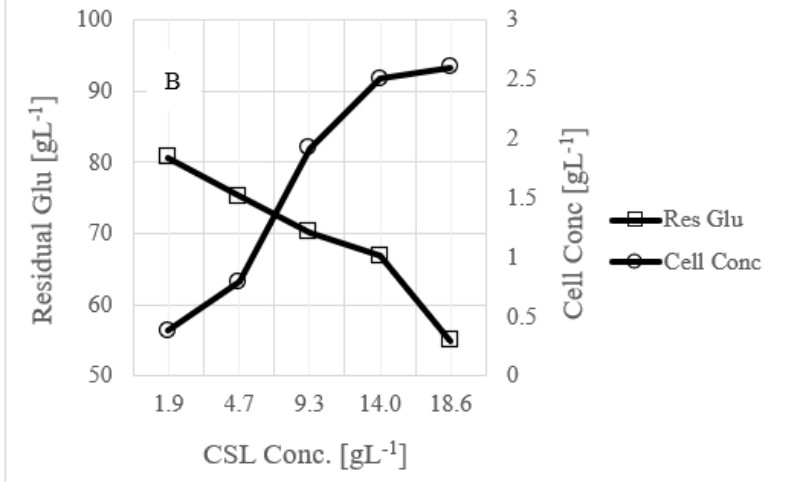

**Figure 4.** *Cont.*

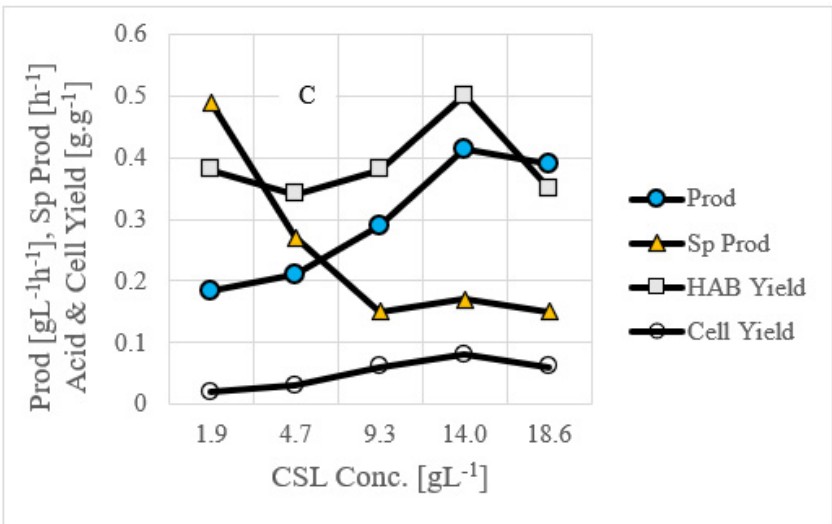

**Figure 4.** Effect of CSL on *C. tyrobutyricum*. (**A**) Production of acids at various CSL concentrations; (**B**) residual glucose and cell concentration; (**C**) productivities, specific productivities, acid yield, and cell yield.

### 3.4. Maximization of Butyric Acid Production in Batch Fermentation

The abovementioned fermentations were run without pH control. It is known that many clostridia regulate their own pH [22]. It was noted that the *C. tyrobutyricum* NRRL 67062 strain did not regulate its pH, and the final pH values were 4.54 to 5.63. These pH values were considered to be low. External pH can affect the cell wall structure and alter cell permeability to ions and other minor metabolites, which negatively affects growth and fermentation. pH also affects the shape of enzymes involved in the fermentation process. In our pH-uncontrolled fermentation (control fermentation), the total acid concentration was observed to be 20.90 g $L^{-1}$, which is comparable to the 20.40 g $L^{-1}$ reported by Wu and Yang [17] at a pH of 5.5.

Hence, fermentations were next run with pH control using the RCM and glucose medium. The purpose of this fermentation was to maximize butyric acid production. The results of this experiment are shown in Figure 5A. The fermentation was ended once the glucose was exhausted. Based on butyric acid inhibition experiments, it was expected that the culture would produce > 50 g $L^{-1}$ of butyric acid. Under pH-controlled conditions, the strain was able to tolerate a higher concentration of acid. At this stage, it is not clear why the strain tolerated a higher concentration of acid in pH-controlled fermentation. It is likely that at an optimum pH, the strain did not experience stress. After 103 h, the culture produced 57.86 g $L^{-1}$ of butyric acid and 12.88 g $L^{-1}$ of acetic acid, thus producing 70.74 g $L^{-1}$ of total acids. The final cell concentration and residual glucose levels were 3.80 and 0.0 g $L^{-1}$, respectively. Based on the consumed glucose, the combined butyric and acetic acid yield was 0.71 g $g^{-1}$, which is remarkably high compared with the yields observed using the RCM in pH-uncontrolled fermentations. In this fermentation, the acid productivity was 0.69 g $L^{-1}$ $h^{-1}$, which was maximum in the present study. In the control fermentation, a productivity of 0.50 g $L^{-1}$ $h^{-1}$ was achieved, which was 71% of that achieved in this pH-controlled fermentation.

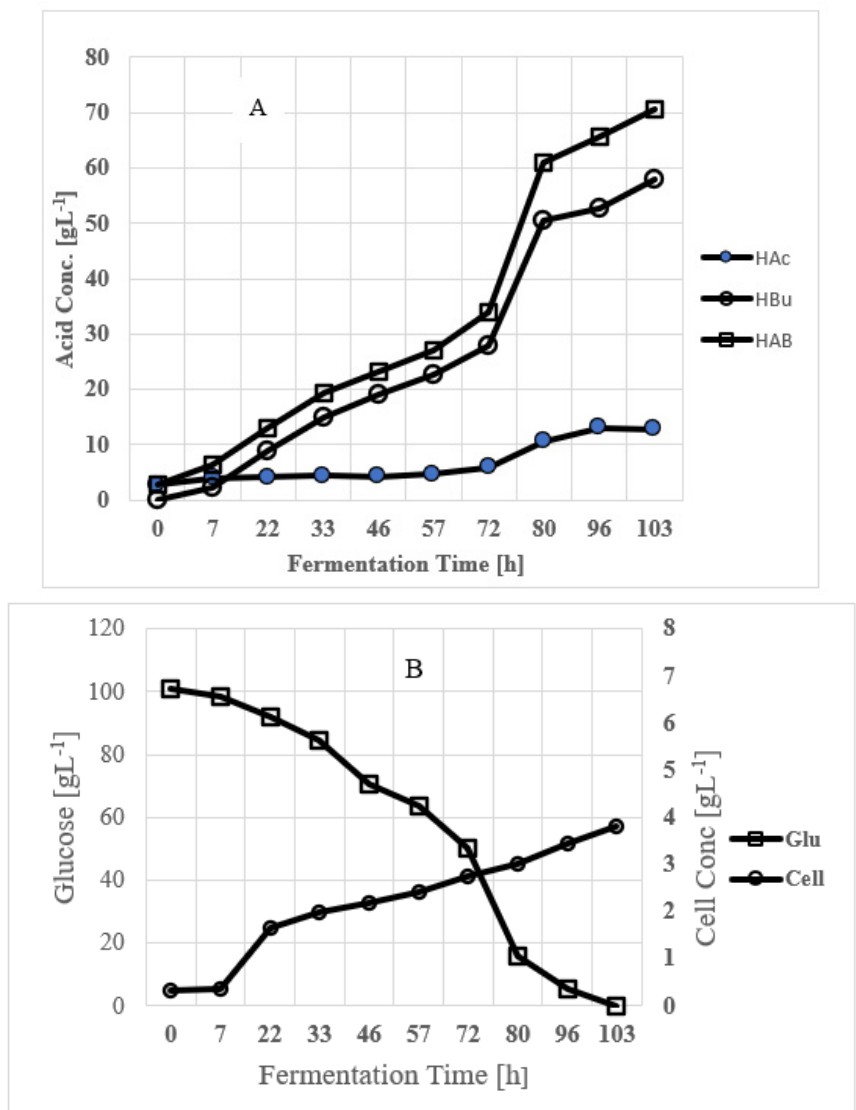

**Figure 5.** Fermentation time profile. (**A**) Acid concentrations. (**B**) Glucose and cell concentrations.

*C. tyrobutyricum* ATCC 25755 is another excellent butyric acid producer that has been reported to achieve 58.80 and 54.79 g $L^{-1}$ butyric acid concentrations [11,27]. Xiao et al. [10] reported the production of 50.37 g $L^{-1}$ of butyric acid in their reactor. Our strain (NRRL 67062) produced 57.86 g $L^{-1}$ of butyric acid and 12.88 g $L^{-1}$ of acetic acid for a total acid production of 70.74 g $L^{-1}$. It is suggested that due to economic and environmental reasons, both butyric and acetic acids should be recovered. The separation of butyric and acetic acids can be performed using salting-out extraction [30]. In our reactor, the culture accumulated 3.80 g $L^{-1}$ of cell mass as opposed to 15.7 g $L^{-1}$ reported by Sjöblom [9]. The higher biomass concentration (15.7 g $L^{-1}$) acted as a drag on organic acid production because more organic carbon was going into forming cell mass. Nevertheless, it may have been the reason for the high productivity achieved in their fermentation. Product or acid yield is another important factor that should be compared. Oh et al. [31], Sjöblom et al. [11], Xiao et al. [10], and Wu and Yang [17] reported acid yields of 0.46, 0.52, 0.38, and 0.45 g $g^{-1}$, respectively. In our fermentation, the total acid yield was 0.71 g $g^{-1}$, which was superior to the yields reported by the abovementioned investigators. The keys to butyric acid's successful fermentation are adequate amount of nutrients and pH control. The microorganism performed better with pH regulation, mild agitation, and anaerobic conditions. It is our aim to mutate the strain (NRRL 67062) to block acetic acid production, which will aid the downstream processing of butyric acid.

In summary, acid production was increased from 20.90 g L$^{-1}$ in the beginning fermentation to 70.74 g L$^{-1}$ by using pH control. This was an increase of 338%. Furthermore, using CSL as a nutrient source paves the way for developing an industrially favorable medium. During these experiments, cell yields were low (0.02–0.08 g g$^{-1}$), which is also beneficial because more sugar is shuttled towards acid production. As originally reported, this strain produced less than 9.87 g L$^{-1}$ of butyric acid [7]. As a result of optimization, butyric acid production was greatly enhanced using this novel strain.

## 4. Conclusions

In a control batch fermentation, *C. tyrobutyricum* NRRL 67062 produced 15.62 g L$^{-1}$ of butyric acid and 5.29 g L$^{-1}$ of acetic acid (total acids of 20.90 g L$^{-1}$) from a 60 g L$^{-1}$ glucose solution. The strain was strongly inhibited above 200 g L$^{-1}$ of glucose. At 60 g L$^{-1}$ of glucose, the productivity of the acids was 0.50 g L$^{-1}$ h$^{-1}$, while it was 0.18 g L$^{-1}$ h$^{-1}$ at 200 g L$^{-1}$ of glucose. As glucose concentration increased to 250 g L$^{-1}$, productivity decreased to 0.16 g L$^{-1}$ h$^{-1}$. The cell growth was not inhibited until 25 g L$^{-1}$ of butyric acid was added to the medium. Butyric acid inhibition and pH control experiments suggested that this strain can tolerate 71 g L$^{-1}$ of total acids (butyric and acetic acids). CSL was found to support cell growth and butyric acid production better than the RCM. Furthermore, the production of butyric acid was scaled up to a 2.5 L bioreactor. In a pH-controlled bioreactor, the culture produced 57.86 g L$^{-1}$ of butyric acid and 12.88 g L$^{-1}$ of acetic acid, thus totaling 70.74 g L$^{-1}$ of acid production. These results suggest that this *C. tyrobutyricum* strain NRRL 67062 has potential commercial interest for the production of butyric acid.

**Author Contributions:** N.Q.—project administration and resources; N.Q. and S.L.—conceptualization. N.Q. performed experiments and wrote the manuscript. S.L. and B.C.S.—review and writing. All authors have read and agreed to the published version of the manuscript.

**Funding:** Financial support was provided by the United States Department of Agriculture, Agricultural Research Service (CRIS Number 5010-41000-189-00D).

**Institutional Review Board Statement:** Not applicable. This study did not involve humans or animals.

**Informed Consent Statement:** Not applicable. This study did not involve humans.

**Data Availability Statement:** All data generated during this study are presented in this article and are available from the corresponding author on request.

**Acknowledgments:** N.Q. would like to thank Bruce S. Dien and Christopher Skory (both of the United States Department of Agriculture, Agricultural Research Service, National Center for Agricultural Utilization Research, Peoria, IL, USA) for critically reading this manuscript and providing highly constructive comments that improved quality of the paper. N.Q. would also like to thank Maria S. Brauer (USDA, ARS, NCAUR, Peoria, IL, USA) for her excellent help with performing experiments and analyzing samples with GC and HPLC. Mention of trade names or commercial products in this article is solely for the purpose of providing scientific information and does not imply recommendation or endorsement by the U.S. Department of Agriculture. USDA is an equal opportunity provider and employer.

**Conflicts of Interest:** The authors declare no conflict of interest.

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
