# Peer review of "Butyric Acid Production by Fermentation: Employing Potential of the Novel Clostridium tyrobutyricum Strain NRRL 67062"

_fermentation, doi:10.3390/fermentation8100491_

Round 1

Reviewer 1 Report

I have carefully revised the study entitled (Butyric Acid Production by Fermentation: Employing Potential of the Novel Clostridium tyrobutyricum strain NRRL 67062). The study is very well with some few primitive experiments on glucose inhibition and butyric acid inhibition. Then one experiment with pH control fermentation. It is well known that for the enhancement of the acid production, pH controlled fermentation is always the best option. However, in reality if the authors could produce only acid in its normal form it would be better for the downstreaming processes.

The scenario and the effective utilization would be better if the authors could optimize the fermentation conditions by the selected strain after the characterization process. Additional experiments are required and the overall experimental design need to be improved.

Some minor comments

Beside, some the authors show some usefulness tables for preparation of their media (Table 1-2), moreover in table 2, what is the concentration of used butyric acid stock.

Abstract, correct (in these studies)

Author Response

Dear Editor:

Please see attached file below. This includes our response to Reviewer I.

Reviewer 2 Report

Although the outcomes of this study are very interesting and could be useful to the scientific community working on volatile fatty acids production, in my opinion some aspects should be improved and more critical discussion of the obtained results is needed before reconsidering its possible publication.   

Specific comments: 

Abstract, Line 16: “To characterize glucose inhibition, sugar concentrations ranging from 60 to 250 g·L−1 were fed to the reactor,” At this point of the paper, the reader do not know how the reactor was operated. Please provide more information of the tests performed. 

Line 38: “including glucose derived from corn, corn, sugarcane molasses,” The word ‘corn’ is repeated two times in this sentence. Please check. 

Line 117: “Clear supernatant was stored at -18 ℃ until analyzed for residual glucose, and acetic and butyric acids.” Please discuss at this point if the organic matter mass balance could be completed with glucose, acetic acid and butyric acids or other organic matter compounds (i.e. other carboxylic acids) could be also produced. 

Line 142: “pH was controlled at 6.5 automatically using 5M NaOH” Please indicate why a pH of 6.5 was selected. 

Line 201: “thus suggesting that this strain can tolerate and produce greater than 50 g·L−1 butyric acid.” Please compare at this point the butyric acid production of this strain when compared with others available in literature to highlight its importance. 

Line 204: “Challenging the culture with butyric acid greater than 25 g·L−1 sharply reduced cell growth.” Although the cell growth was reduced the residual glucose concentration also decreased. Please discuss this fact in the text and why it took place. 

Line 222: “At the lowest RCM concentration tested (3.80 g·L−1), the yields were only 3.41 g·L−1 butyric acid and 2.71 g·L−1 acetic acid.” Do the authors refer to ‘yields’ or maximum concentrations achieved? Please check. 

Line 262: “The above fermentations were run without pH control.” If available, please clearly indicate in the previous tests the final pH achieved since the fermentation yield and concentration could be highly affected by pH. 

Please enrich the critical discussion of the paper by comparing the acetic and butyric acids yield and distribution with others obtained in literature.

Author Response

Dear Editor:

Please find a file that contains our response to the reviewer II comments.

Reviewer 3 Report

In this study, Clostridium tyrobutyricum NRRL 67062 was used to produce butyric and acetic acids using glucose and then with corn steep liquor to improve the productivity. In addition, the strain was studied for glucose and butyric acid inhibition. The following queries should be addressed before considering further processing.

Line #39: corn, corn, sugarcane molasses – Remove the repetitive corn.

No need to mention the full name of the genus after first mentioning it in the manuscript (C. tyrobutyricum is fine).

In 2.1, include the OD of inoculum I and II.

Check the units of acid and cell yield in Fig. 1B

Include the possible justification for increasing butyric acid tolerance under controlled pH experiments.

Some recent references could be included in the manuscript.

Author Response

Please find attached a file that contains our response to the reviewer III comments.

Reviewer 4 Report

Although the production of organic acids through the microbial route is of commercial significance, the major aim of this study, the authors would have provided a strong discussion stating the novelty of the study. With the provided comments, the manuscript needs significant revision, in my opinion in the current format the manuscript is not acceptable for publication

Author Response

Dear Editor:

Please see our response to the reviewer 4 comments. File attached. Thank you for getting this Ms reviewed.

Round 2

Reviewer 1 Report

The authors have addressed the raised comments. but the optimization results should be indicated and described correctly even in supplementary files

Reviewer 4 Report

Authors has addressed the comments and suggestions provided, in my opinion the manuscript can be accepted for the publication.